# Estimating the Direct Effect between Dietary Macronutrients and Cardiometabolic Disease, Accounting for Mediation by Adiposity and Physical Activity

**DOI:** 10.3390/nu14061218

**Published:** 2022-03-13

**Authors:** Hugo Pomares-Millan, Naeimeh Atabaki-Pasdar, Daniel Coral, Ingegerd Johansson, Giuseppe N. Giordano, Paul W. Franks

**Affiliations:** 1Lund University Diabetes Centre, Department of Clinical Sciences, Lund University, Skåne University Hospital, 21428 Malmö, Sweden; hugo.pomares-millan@med.lu.se (H.P.-M.); naeimeh.atabaki_pasdar@med.lu.se (N.A.-P.); daniel.coral@med.lu.se (D.C.); giuseppe.giordano@med.lu.se (G.N.G.); 2Department of Public Health and Clinical Medicine, Umeå University, 90187 Umeå, Sweden; ingegerd.johansson@umu.se; 3Department of Nutrition, Harvard T.H. Chan School of Public Health, Boston, MA 02115, USA

**Keywords:** macronutrient intake, mediation, causal inference, cardiometabolic risk, cardiovascular disease, adiposity, physical activity

## Abstract

Assessing the causal effects of individual dietary macronutrients and cardiometabolic disease is challenging because distinguish direct effects from those mediated or confounded by other factors is difficult. To estimate these effects, intake of protein, carbohydrate, sugar, fat, and its subtypes were obtained using food frequency data derived from a Swedish population-based cohort (n~60,000). Data on clinical outcomes (i.e., type 2 diabetes (T2D) and cardiovascular disease (CVD) incidence) were obtained by linking health registry data. We assessed the magnitude of direct and mediated effects of diet, adiposity and physical activity on T2D and CVD using structural equation modelling (SEM). To strengthen causal inference, we used Mendelian randomization (MR) to model macronutrient intake exposures against clinical outcomes. We identified likely causal effects of genetically predicted carbohydrate intake (including sugar intake) and T2D, independent of adiposity and physical activity. Pairwise, serial- and parallel-mediational configurations yielded similar results. In the integrative genomic analyses, the candidate causal variant localized to the established T2D gene *TCF7L2*. These findings may be informative when considering which dietary modifications included in nutritional guidelines are most likely to elicit health-promoting effects.

## 1. Introduction

Global patterns of food consumption and energy expenditure have changed drastically in recent decades. Increased sedentary behavior, coupled with the availability of cheap, energy-dense foods, has led to the rapid rise in overweight and obesity worldwide [1]. Excess weight (i.e., body mass index (BMI) > 25 kg/m^2^) is one precursor to type 2 diabetes (T2D) and cardiovascular disease (CVD). Hence, an imbalance between energy intake, physical activity and lifestyle behaviors has a major impact on BMI, CVD and T2D risk. Indeed, the Global Burden of Disease Study 2017 reported that dietary risk accounted for 11 million deaths and 255 million disability-adjusted life year (DALYs) in adults [2].

Recent studies have revealed genetic variants associated with food preferences, dietary patterns and food intake [3,4,5,6,7]. Among those macronutrients ingested, total or specific fats and carbohydrates have been associated with obesity, CVD and T2D, yet controversy remains about whether it is energy density that mediates such associations or if single nutrients (e.g., saturated fat, fructose) increase risk of disease [8]. Clinical trials have indicated that macronutrients might influence glucose metabolism; for example, as part of a lifestyle intervention, a low-energy, low-carbohydrate diet reduced T2D risk [9,10]. Whilst it is plausible that each nutrient could affect disease risk, some might be of greater relevance.

Understanding the causal role of each macronutrient, therefore, could elucidate pathways for more precise dietary intervention strategies [11]. We sought to disentangle the causal role of macronutrients through an integrative analysis using Mendelian randomization (MR) and colocalization obtained through published genome-wide association studies (GWAS) of T2D and CVD. Moreover, we characterize the direct and indirect effects of mediators (i.e., adiposity and physical activity (PA)) on metabolic traits, such as plasma lipids, blood sugar and cardiometabolic disease.

## 2. Materials and Methods

### 2.1. Study Design and Population

The Northern Sweden Diet Database (NSDD) contains data from participants collected within the Västerbotten Health Survey (VHU) [12]. Briefly, VHU is an ongoing, prospective, population-based cohort study started in 1985, where adult residents in the county of Västerbotten in Northern Sweden have been invited to a health examination at 40, 50 and 60 years of age (<1% of 30-year-olds were included initially, then discontinued). For this study, participants screened between 1991 and 2016 were eligible, as they had undergone an extensive health examination by trained nurses and family physicians at their local primary care center, including anthropometry, blood lipids and glucose levels before and after a 75 g oral glucose load, and completed surveys, i.e., food frequency questionnaire (FFQ), socio-economic and lifestyle conditions. Values outside normal ranges suggested by VHU data managers were considered outliers and excluded (see Appendix A). The study protocol and data handling procedures were approved by the Regional Ethical Review Board of Northern Sweden, Umeå, and written informed consent was obtained from all study participants.

### 2.2. Exposure, Mediator and Outcome Measures

Exposure data were derived for participants who completed the FFQ. Two versions were used during the study: a long version (84 items) and a shortened version (64–66 items). The FFQs have been validated against repeated 24 h dietary records and/or biological markers [13]. Daily energy intake and macronutrient subtypes were calculated for each participant from the food composition database provided by the National Food Agency of Sweden (www.livsmedelsverket.se/en/foodand-content/naringsamnen/livsmedelsdatabasen/; accessed 25 June 2021). This included proteins (animal- and plant-based), carbohydrates and added sugar, the latter being estimated by adding all sucrose and monosaccharides intake minus sugars from fruits and vegetables. Total sugar was further calculated as the sum of all monosaccharides and disaccharides in diet. Saturated, trans-and total fat were also obtained per participant. The macronutrient percentage of energy intake (E%) was calculated by multiplying intake by the metabolizable energy conversion factors and dividing this by total energy intake (TEI) [14]. Those that reported taking dietary supplements or vitamins in the last 14 days were not included.

Since fats, proteins and carbohydrates are rarely consumed in isolation, we added the micronutrients queried from the FFQ and obtained nutrient patterns through principal component (PC) analysis to represent a comprehensive characterization of diet in a real-world setting.

As mediators, adiposity was defined as body mass index (BMI), calculated as body weight in kg (using a calibrated weighing scale) divided by height in m^2^, obtained from participants wearing light clothes and no shoes. For physical activity (PA), we calculated a PA index, ranging from 1 = inactive to 4 = active, as described elsewhere [15]. We further included the ‘exercise in leisure time’ variable, reported in five different ordered categories ranging from (1 = never exercise to 5 = more than three times/week). Both were treated as continuous in analyses.

The primary outcomes (T2D and CVD), expressed as binary variables, were obtained through record linkage to the health databases of the National Board of Health and Welfare in Sweden (www.socialstyrelsen.se/register; accessed 25 June 2021). Clinical endpoints were retrieved using ICD-9 code 250 and ICD-10 codes E11.0–E11.9 for T2D. For the composite CVD outcome, ICD-9 code 410 and ICD-10 code I21 were applied for MI. For stroke cases, ICD-9 codes 430, 431 and 433–436 and ICD-10 codes I60, I61, I63 and I64 were used. Secondary outcomes were lipid traits (i.e., high- and low-density lipoprotein (HDL-C, LDL-C, respectively), total cholesterol (TC) and triglycerides (TG)). Glycemic traits included fasting glucose (FG) and two-hour glucose (2 h glucose). For FG, blood was drawn after overnight or 4 h fasting; for 2 h glucose, a blood sample was drawn two hours after the administration of a 75 g oral glucose load, then measured using a Reflotron bench-top analyzer (Roche Diagnostics Scandinavia AB). HDL-C was only measured in a subgroup of participants (*n* = 23,581) and LDL-C was obtained using the Friedewald formula [16]. TG and TC levels were analyzed using standardized chemical analysis [12]. Validated conversion equations were used to adjust blood lipid measurements taken before and after September 2009 [17]. For participants on lipid lowering medication, lipid levels were corrected by adding published constants (+0.208 mmol/L for TG, +1.347 mmol/L for TC, −0.060 mmol/L for HDL-C, +1.290 mmol/L for LDL-C), as recommended elsewhere [18].

### 2.3. Statistical Analysis

The distribution of all continuous explanatory variables was assessed for normality. A constant (0.1) was added to all dietary variables prior to log-transforming to correct skewness. We retrieved complete cases for glycemic (*n* = 55,613) and lipid models (*n* = 23,581). Mediation models were employed to decompose total effects into direct and indirect effects [19]. We used structural equation modelling (SEM) to study the extent to which PA and BMI influenced associations between macronutrient intake and changes in T2D and CVD status, as well as lipid and glycemic traits. In mediation analysis, a pathway of relationships between variables (i.e., exposure, mediator and outcome) can be modelled using generalized linear regression equations according to a prespecified configuration [20]; these analyses also allow covariance between variables to be determined (see below). For indirect pathways, the two hypothesized mediators of macronutrient intake (PA and BMI) were fitted into pairwise models (Figure 1A) [21].

Next, we fitted parallel mediation models (i.e., exposure → PA → outcome and exposure → BMI → outcome) (Figure 1B) [20] and, given PA and BMI are often correlated, serial mediation models were also tested (exposure → PA → BMI → outcome in Figure 1C). Estimates and standard errors (SE) were obtained through bootstrapping (5000 draws), as recommended elsewhere [22]. To represent real-world dietary habits, all raw nutrient variables were adjusted for TEI using the residual method [23], then centered and scaled to obtain PCs of dietary patterns.

#### 2.3.1. Mediation Analysis

Overall, the mediation analysis is constructed using three linear equations:*Y* = *i*1 + *cX* + *ϵ*1(1)
*Y* = *i*2 + *c′X* + *bM* + *ϵ*2(2)
*M* = *i*3 + *aX* + *ϵ*3(3)
where *i*1, *i*2 and *i*3 are intercepts, *Y* is the outcome, *X* is the explanatory variable, *M* is the mediator and *ϵ* represents the error term. Thus, under the sequential ignorability assumption [24], the model equation can be expressed as:*Y* = *i*2 + *bi*3 + (*c′* + *ab*) *X* + *ϵ*2 + *bϵ*3(4)

For pathways *a*, *b*, *c′*, the following models were fitted: (i) a linear regression assessing the association between each macronutrient (or PC) and the mediators BMI and/or PA, either in serial or in parallel form (pathway *a*); (ii) a linear or logistic regression between mediators BMI and/or PA and the outcome, adjusted for changes in macronutrient intake (pathway *b*); (iii) linear or logistic regression assessing associations between macronutrient intake (or PC) and outcomes, having adjusted for mediators (pathway *c′*). The indirect effect (*a* × *b*) was quantified as the effect of the mediators (BMI and PA), and the total effect by the sum of indirect and direct effects (*c′ + ab* in Equation (4)). To assess multicollinearity between variables, the variance inflation factor (VIF) was calculated (variables > 10 were removed). All models were adjusted for putative confounders for each outcome (i.e., age, sex, education, TEI, portion size of potatoes, meat and vegetables, fiber intake (g/day), and alcohol intake (g/day)). For the CVD composite outcome, we further adjusted for tobacco use. Statistical significance was *p* < 0.05 (two-tailed test); in pairwise analyses, a false discovery rate (FDR) correction was set at P_FDR_ < 0.05 under the Benjamini–Hochberg procedure [25].

#### 2.3.2. Two-Sample Mendelian Randomization and Bayesian Colocalization

Genetic variants, used here as instrumental variables (IVs) for dietary intake, are randomly assorted during conception [26] and, thus, can be employed for causal inference. For IVs to be valid, they should be associated with the exposure, unrelated to confounders of the exposure–outcome association; they should also affect the outcome only via the exposure (Figure 1D). We assessed the causal impact of dietary carbohydrates, sugars, fat and protein intake with glycemic and lipid traits, T2D and CVD (i.e., stroke and CHD), in a two-sample MR framework (2SMR). The SNPs for exposure data were retrieved from public GWAS summary data from Meddens et al. [5], which were derived from the Social Science Genetic Association Consortium (SSGAC) in 268,922 European ancestry participants. A more detailed description of the dataset is available in their website (https://www.thessgac.org/data; accessed 1 July 2021). Briefly, all dietary intake data were obtained through self-reported food frequency questionnaires and single 24 h diet recalls (only for UK Biobank), and macronutrients were reported as % of energy intake (E%). Owing to the low number of GWAS-significant SNPs in the exposures (6 for fat, 7 for protein, 13 and 10 for carbohydrate and sugar intake, respectively), we relaxed the GWAS threshold to *p*-value < 
$5\times 10^{-6}$
. Further, proxies were used if genetic variants were in linkage disequilibrium (LD) at 
$r^{2}$
 ≥ 0.8 in any of the two-samples. To minimize correlations between the IVs, we performed LD-clumping (where SNPs with lowest *p*-value are retained) restricted to 
$r^{2}$
 < 0.2 in a 1000 kb window for the final sets. To disentangle the effect of carbohydrates from sugar (considered a subcomponent in the original GWAS [5]), we combined the significant sugar- and carbohydrate-associated SNPs (*n* = 79) at the set threshold (*p*-value < 
$5\times 10^{-6}$
). Those overlapping (*n* = 4) were removed to avoid pleiotropy. To construct the IVs for the outcome variables, we used GWAS available in European ancestry populations. CAD GWAS summary statistics were derived from the Coronary Artery Disease (C4D) Genetics consortium (CARDIoGRAMplusC4D) [27], which included 60,801 cases of CAD and 123,504 controls. For stroke, summary statistics were obtained from the MEGASTROKE consortium, which includes 40,585 cases and 406,111 controls [28]. For T2D, we obtained the unadjusted and BMI-adjusted summary statistics, which include 48,286 cases and 250,671 controls from the DIAGRAM consortium [29]. We used data derived from the MAGIC consortium for fasting [30] and 2 h glucose [31]. For lipid traits, we used data derived from a recent secondary analysis in UK Biobank for TG, HDL-C, and LDL-C [32]. For TC, we used data from a recent GWAS [33]. Characteristics of all GWAS utilized in this study are in Appendix A.

We used the inverse variance weighted (IVW) method for our main analysis to estimate the effects of the IVs. Moreover, we used MR-Egger and weighted median estimators to address consistency. As the number of instruments was expected to be low, we used the median F-statistic to measure the IV strength. We further employed the robust adjusted profile score (MR-RAPS) method, by weighing each variant for the effect and precision of the SNP-exposure association, as recommended when using weaker instruments (i.e., below the conventional GWAS threshold [34]). To quantify heterogeneity, bias from horizontal pleiotropy and outliers, we estimated the Cochran’s *Q* statistic for MR-Egger and IVW, and the MR Pleiotropy Residual Sum and Outlier (MR-PRESSO) global test at *p* level of >0.05 [35]. Exposure and outcome data were harmonized to ensure alleles were aligned, with ambiguous and/or palindromic variants being removed. In addition, we estimated the potential of sample overlap according to Burgess et al. [36] (Appendix A). We also performed a leave-one-out sensitivity analysis to assess the impact of each SNP (Appendix A). To identify shared causal pathways among traits, we employed the Hypothesis Prioritization for multi-trait Colocalization (HyPrColoc) algorithm [37], which identifies genome-wide regions with evidence of shared variants (putative of a causal pathway) across traits (Appendix A). All statistical analyses were performed with R version 3.6.2. Mediation analyses were performed with the ‘mediation’ [21] and ‘lavaan’ R packages [38]. Two-sample MR analysis was conducted using ‘TwoSampleMR’ [39] and ‘MendelianRandomization’ [40]. Colocalization was performed with the ‘HyPrColoc′ [37] and ‘coloc’ R packages [41], and PC analysis was visualized with ‘PCATools’, ‘ComplexHeatmap’.

## 3. Results

Data from a total of 63,862 participants were analyzed. The mean (SD) age of the cohort was 46.5 (8.37) years and 50.3% were female. The means (SD) of glycemic and lipid traits were FG 5.44 (0.93) mmol/L; 2 h glucose 6.55 (1.53) mmol/L; TC 5.39 (1.09) mmol/L, LDL-C 3.59 (1.06) mmol/L and HDL-C 1.37 (0.46) mmol/L, and the median TG was 1.40 (0.81) mmol/L (see Appendix A). Genetic correlations were computed using LD Score Regression [42] for traits for which GWAS summary statistics were available, and Pearson’s pairwise correlations among mediators and outcomes are shown in Appendix A. For PC analysis, we selected the top three PCs that explained >52% of the total variance (Appendix A) to maintain distinctive dietary patterns. The ten variables contributing the most to the top three PCs are plotted in Appendix A. From these, ‘polyunsaturated fat’ and ‘total fat’ were observed in PC1 and PC3. The variable with the largest loading value for PC 1 was ‘fiber’, for PC 2 it was ‘sucrose’ and for PC 3 ‘polyunsaturated fat’. The correlation among traits, nutrients and PCs are shown in Figure 2.

### 3.1. Mediation Analysis

The direct and indirect effects for each macronutrient (or PC)–mediator associations are depicted in Figure 3 and summarized in Appendix A. In parallel and serial mediation models, given that we were mainly interested in the direct effect of our exposures, we compared partially and fully mediated nested models (i.e., Figure 1B,C with and without pathway *c′*, respectively) using the chi-squared difference test [43]. The bootstrapped direct and indirect effect estimates, standard errors, and fit indices for parallel and serial mediation models are summarized in Appendix A.

For those macronutrients that remained significant after correction (P_FDR_ < 0.05) with glycaemic traits, i.e., FG, we identified nine direct effects (Appendix A)—these included added sugar, total sugar, trans-fat, total carbohydrates with positive direction, and with negative effects—saturated and total fat. For 2 h glucose, negative direct effects were observed for saturated, trans-, and total fat (Appendix A). Moreover, either in serial or in parallel form, the fully mediated models were not statistically different from the partially mediated model (Appendix A).

With respect to lipids, there were four direct effects for HDL-C, these consisted of total carbohydrates, added and total sugar with negative direction, and total fat with positive effects; all macronutrients in their fully mediated models were statistically different from the partially mediated model, favoring the latter. Three direct effects from total fat and plant-based protein (negative) and trans-fat (positive) were observed for LDL-C; For TC, plant-based proteins and total fat (negative), trans-and saturated fat (positive) had evidence of direct effect. Only total fat and its subtypes had negative direct effects on TG (Appendix A).

For T2D, total carbohydrates and trans-fat had positive significant effects, whilst saturated and total fat had an opposite effect; the partially mediated models were significantly different from the fully mediated models, favoring the former (Appendix A). With respect to CVD, total protein intake was the only macronutrient without significant direct and total effects, irrespective of mediational configuration (Appendix A).

### 3.2. MR Causal Effects

In MR analyses, carbohydrate intake was associated with T2D per E% unit increase: OR_IVW_ 0.1 (95% CI: 0.013, 0.71; *p* = 0.02); however, the MR-Egger estimate was not significant, yet when using T2D adjusted for BMI (T2DadjBMI), the effect decreased to OR_IVW_ 0.47 (95% CI: 0.3, 0.75; *p* = 0.001) with *β*_MR-RAPS_ −0.82 (se 0.3; *p* = 0.004) and no evidence of pleiotropy P_MR-PRESSO_ = 0.43 (Figure 4 and Table 1).

Regarding the effect of carbohydrate intake on lipid levels, TC, LDL-C and TG per E% unit change 9*β*_IVW_ 0.32 (95% CI: 0.02, 0.63; *p* = 0.03, *β*_IVW_ 0.44 (95% CI: 0.05, 0.82; *p* = 0.03), and *β*_IVW_ 0.1 (95% CI: 0.01, 0.2; *p* = 0.03), respectively), yet there was evidence of pleiotropy. For the carbohydrate adjusted for sugar intake instrument (6 SNPs instrumentalized) per E% unit change and T2D, the effect estimate was *β*_IVW_ 0.09 (95% CI: −7.7, 7.9; *p* = 0.9), and not significant MR-Egger and MR-RAPS models (Appendix A). Moreover, for fat when undertaking MR-Egger, there were no significant associations with any outcome (Appendix A).

## 4. Discussion

We report a comprehensive analysis investigating mediational and causal effects of macronutrient intake and cardiometabolic traits and diseases in >60,000 Swedish participants. To our knowledge, this is the first study reporting the likely causal role of macronutrient intake and the risk of cardiometabolic disease, triangulating evidence from observational and genetic studies. Implications of our findings indicate carbohydrate intake (with predominance of fiber) is likely followed by reduction in T2D risk. By contrast, sugar intake likely raises T2D risk. Due to the modest magnitude of observed effects, it is unlikely to prove a useful target when intervening only through diet for disease prevention. These findings reinforce the notion that complex carbohydrates may be recommended in dietary modifications, alongside other lifestyle changes, to lower individuals’ risk of T2D.

The apparent protective effects of dietary carbohydrates in T2D suggests that the quality of carbohydrate is key in T2D prevention. Previous observational studies indicate that associations with T2D can vary according to the carbohydrate type [44], i.e., fiber (sourced from fruits, vegetables or cereals) had a protective effect [45], whereas starch had deleterious effects [46]. In our MR analyses, it was not possible to interrogate carbohydrate or sugar subtypes. Mechanistic studies show that carbohydrate metabolism is heavily dependent on insulin action. However, the fiber effect is believed to be secondary to the transformation to β-glucans, a water-soluble gel-forming substance that decreases surface of exposure in the small intestine, delaying the gut absorption of glucose and reducing postprandial plasma glucose [47]. Moreover, dietary fiber has been associated with lower energy intake and increased satiety [48]. The most probable causal locus, *TCF7L*2, is an established T2D-associated gene [49] which appears to interact with intake of dietary fiber [50], fat [51] and whole grains [52]. Nevertheless, *TCF7L*2’s mechanisms of action, especially in the context of interactions with dietary factors, remains poorly defined. Recent evidence suggests a key role of glucagon-like peptide 1 (GLP-1), secreted after meal ingestion [53], or serotonin [54]. More recent findings from pooled clinical trials in T2D have emphasized the role of gut microbiome in the transformation of fiber-rich foods and glycemic markers [55]. With respect to lipid markers, our observational findings are in line with those reported in previous studies [56], where carbohydrate intake has been linked to LDL-C, HDL-C, TC and TG. Yet, in our MR findings, there was no evidence of causality. For protein intake, studies evaluating protein subtypes have shown a protective effect of plant-based proteins against CVD [57]; conversely, proteins from animal sources increased CVD risk [58]. It was not possible to interrogate protein subtypes with MR; yet this source of heterogeneity may explain the observed pleiotropy.

Our study had limitations. Firstly, although SEM allows direct effect modelling, and despite the multiple configurations explored, our hypothesized models do not cover all possible pathways. Moreover, conditioning on a potential mediator or a shared outcome can induce bias. Secondly, inconsistent mediation (positive direct and negative indirect effects or vice versa) was observed for some of the pairwise associations between the independent and mediating variable, suggesting the mediator was not a significant predictor of the outcome when including both. Thirdly, in MR analysis, horizontal pleiotropy and population stratification were addressed using conventional statistical solutions, yet bias cannot be completely ruled out given the paucity of variants available to construct the IVs and other genetically driven individual features (e.g., microbiome composition) [59] may influence the observed associations, moreover, evidence of weak instrument bias may still be present, as indicated by the F-statistics. Fourth, not all macro- and micronutrients (including subtypes) had corresponding genetic instruments; thus, we cannot assess with sufficient granularity the causal effect of single-nutrient intake. Further caveats are that dietary patterns seldomly remain the same over the life course, in contrast to a person’s nuclear DNA variation, which is fixed at conception. Moreover, observational FFQ data were self-reported and estimated effects may be larger than those observed in a real-world setting. Thus, we cannot rule out residual confounding. Another consideration is the generalizability of our findings. Given that the populations included for mediation analysis and MR were predominantly of European ancestry, our findings may not generalize to other ethnicities. Nevertheless, consistent findings across and within methods help ensure detected relationships are robust to confounding and bias, thereby minimizing false positive association, and support the contemporary view that carbohydrates play a causal role in T2D beyond PA and adiposity.

## 5. Conclusions

Our analyses highlight the direct effect of carbohydrate intake in T2D risk, helping to quantify the role of higher-quality carbohydrates (which lower risk). These findings warrant confirmation through clinical trials; however, they may enhance current nutritional guidelines by helping distinguish the dietary factors that are likely to be causal from those that are mostly mediated.

## Figures and Tables

**Figure 1 nutrients-14-01218-f001:**
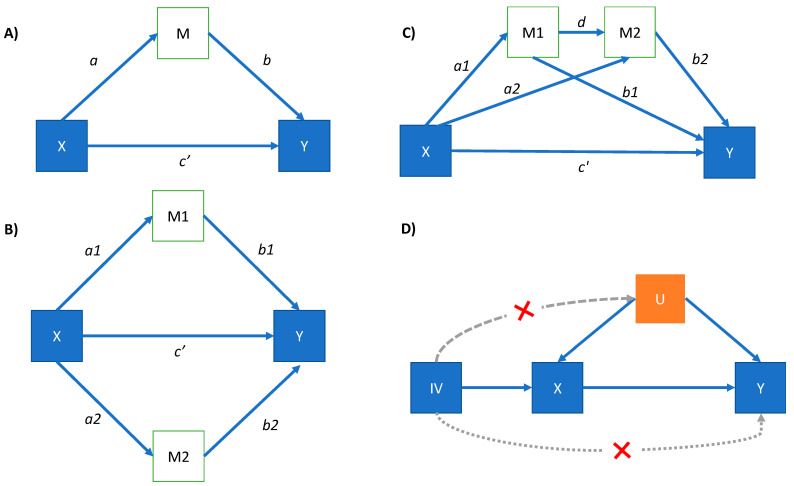
Hypothetical directed acyclic graph models. (**A**) Pairwise mediation model; (**B**) Parallel mediation model; (**C**) Serial mediation model; (**D**) Mendelian randomization model. X: independent variable; M: mediator; Y: outcome; IV: instrumental variable; U: confounding. SEM Pathways: *a* is the coefficient of the effect of X on M; *a*1 and *a*2 are coefficient effects between X and mediators 1 (M1) and 2 (M2), respectively. *b* is the effect of M on Y adjusting for the explanatory variable; *b*1 and *b*2 are coefficient effects between mediators 1 (M1) and 2 (M2), and Y, respectively; *c′* is the coefficient of the effect of X on Y adjusting for M (direct effect), and *d* is the coefficient effect between mediators. For (**D**) in MR, the IV must not be related to confounders (dotted line) of the exposure–outcome association and affect the outcome only via the exposure and not through another via (dotted lines).

**Figure 2 nutrients-14-01218-f002:**
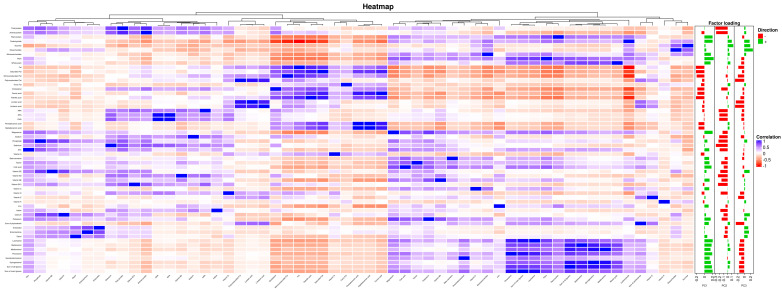
Heatmap of FFQ with 57 items and 3 PCA factor loadings. Correlation key: blue represents positive Pearson’s correlations and red represents negative Pearson’s correlations. Direction key: red represents a negative direction and green represents a positive direction, large loadings (bars) mean that a variable has greater effects on the principal component.

**Figure 3 nutrients-14-01218-f003:**
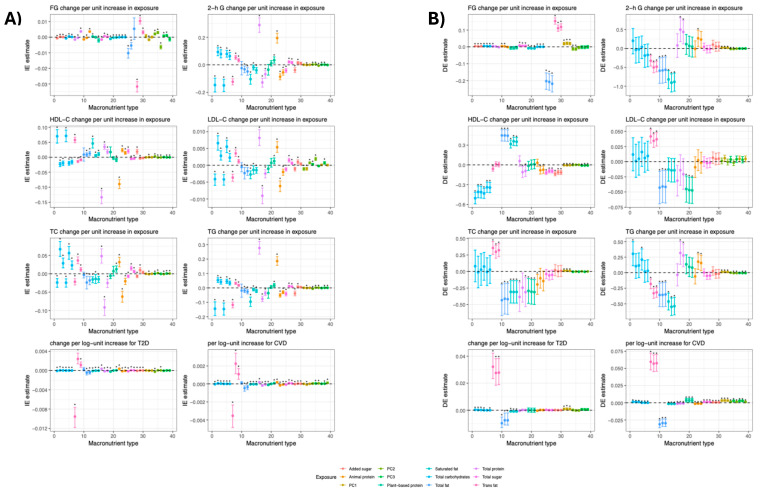
Direct and indirect estimates between macronutrients and outcome in pairwise mediation analysis with body mass index, 5-level physical activity and physical activity index as mediators. Macronutrients are organized on the x-axis in colour codes, ordered consecutively (from left to right) for body mass index, 5-level physical activity and physical activity index. Data are presented as (**A**) indirect and (**B**) direct estimates and 95% confidence intervals; Indirect effect is the estimated average increase in the dependent variable as a result of the mediators; (*) significant after FDR correction at *p* < 0.05; HDL-C, LDL-C: high- and low-density lipoprotein, respectively; TC: total cholesterol; TG: triglycerides; FG: fasting glucose; 2-h G: two-hour glucose; Units: FG mmol/L; 2-h G mmol/L; TC mmol/L; LDL-C mmol/L; HDL-C mmol/L; TG mmol/L; For T2D and CVD, the unit increase corresponds to the probability.

**Figure 4 nutrients-14-01218-f004:**
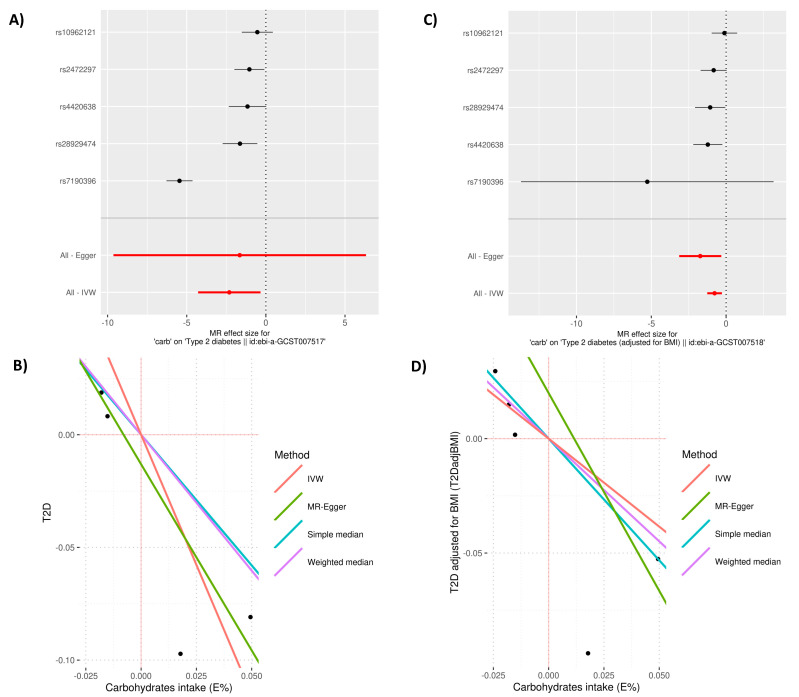
Forest plot of 5-SNP instrument and scatter plot of SNP effects on exposures versus outcomes using different MR methods. For the forest plot, effect size and 95% confidence intervals (standard deviation (SD) change) of the impact of carbohydrates intake SNPs. (**A**,**B**) correspond to carbohydrates (E%)→T2D; (**C**,**D**) correspond to carbohydrates (E%)→T2D adjusted for BMI (T2DadjBMI).

**Table 1 nutrients-14-01218-t001:** Two-sample MR exposure–outcome associations per macronutrient type.

				IVW						MR-Egger						MR-PRESSO		MR-RAPS		
Exposure	Outcome	Number of SNPs	F	β	95% CI		*p*-Value	Q Statistic	*p*-Value	β	95% CI		*p*-Value	Q Statistic	*p*-Value	Global Test *p*-Value	Distortion Test *p*-Value	β	βSE	*p*-Value
**Sugar**	FG	26	5	−0.09	−0.18	0.01	0.07	24.48	0.18	−0.11	−0.5	0.29	0.6	24.39	0.14	0.17	-	−0.08	0.05	0.15
	2 h glucose	24	5	−0.07	−0.54	0.41	0.79	16.02	0.85	−0.93	−2.9	1.05	0.36	15.24	0.85	-	-	−0.04	0.25	0.86
	HDL-C	40	4	−0.05	−0.25	0.16	0.66	1265.66	3.62 × 10^−240^	−0.21	−0.89	0.48	0.55	1253.76	2.02 × 10^−238^	<1 × 10^−4^	0.6	−0.04	0.06	0.49
	LDL-C	40	4	0.43	0.05	0.82	0.03	3787.17	-	1.06	−0.21	2.33	0.1	3695.86	-	<1 × 10^−4^	<1 × 10^−4^	0.1	0.08	0.16
	TC	40	4	0.32	0.004	0.64	0.05	673.38	1.29 × 10^−116^	0.88	−0.17	1.91	0.1	657.34	6.04 × 10^−114^	<1 × 10^−4^	<1 × 10^−4^	0.12	0.09	0.17
	TG	40	4	0.17	0.02	0.32	0.03	619.14	1.65 × 10^−105^	0.32	−0.17	0.82	0.2	611	1.87 × 10^−104^	<1 × 10^−4^	9.00 × 10^−4^	0.05	0.05	0.28
	T2D	2	7	0.04	0.002	0.84	0.04	22.32	2.30 × 10^−6^	-	-	-	-	-	-	-	-	−2.42	1.3	0.06
	Stroke	0	-	-	-	-	-	-	-	-	-	-	-	-	-	-	-	-	-	-
	* T2D	2	6	3.9	0.02	969.07	0.02	0.24	0.63	-	-	-	-	-	-	-	-	1.36	2.91	0.64
	CHD	40	4	1.15	0.85	1.56	0.47	89.69	7.20 × 10^−6^	2.02	0.64	6.32	0.32	87.4	9.20 × 10^−6^	<1 × 10^−4^	<1 × 10^−4^	0.09	0.14	0.51
**Fat**	FG	22	5	0.02	−0.07	0.11	0.68	28.16	0.14	−0.23	−0.49	0.04	0.09	24.9	0.21	0.15	-	0.01	0.06	0.92
	2h glucose	22	5	0.09	−0.43	0.62	0.73	18.73	0.6	−0.31	−2.27	1.65	0.76	18.37	0.56	0.62	-	0.17	0.29	0.56
	HDL-C	34	5	−0.16	−0.34	0.02	0.09	696.01	3.65 × 10^−125^	−0.07	−0.53	0.38	0.75	691.01	8.50 × 10^−125^	<1 × 10^−4^	<1 × 10^−4^	−0.01	0.05	0.76
	LDL-C	34	5	−0.38	−0.79	0.04	0.08	3012.02	-	−0.82	−1.84	0.19	0.11	2940.35	-	<1 × 10^−4^	<1 × 10^−4^	−0.19	0.09	0.05
	TC	34	5	−0.28	−0.62	0.06	0.11	550.56	3.74 × 10^−95^	−0.62	−1.44	0.21	0.15	538.59	2.56 × 10^−93^	<1 × 10^−4^	<1 × 10^−4^	−0.16	0.1	0.11
	TG	34	5	0.11	−0.22	0.43	0.51	2018.29	-	−0.13	−0.92	0.67	0.75	1995.28	-	<1 × 10^−4^	3.00 × 10^−4^	0.04	0.06	0.57
	T2D	5	13	2.91	0.47	17.81	0.25	90	-	0.05	1.17 × 10^−4^	22.07	0.34	55.78	-	2.00 × 10^−4^	<1 × 10^−4^	−0.06	0.67	0.93
	** Stroke	1	-	0.92	0.52	1.63	0.78	-	-	-	-	-	-	-	-	-	-	−0.08	0.3	0.79
	* T2D	5	13	0.94	0.59	1.51	0.81	5.11	0.28	0.77	0.1	5.83	0.82	5.04	0.17	0.55	-	−0.06	0.22	0.77
	CHD	31	5	0.81	0.58	1.12	0.29	81.71	1.10 × 10^−6^	0.69	0.36	1.32	0.21	80.83	9.00 × 10^−7^	<1 × 10^−4^	0.82	−0.21	0.14	0.12
**Carbohydrates**	FG	28	5	−0.07	−0.17	0.03	0.16	44.25	0.02	−0.12	−0.59	0.35	0.61	44.16	0.01	0.02	-	−0.13	0.06	0.02
	2h glucose	31	5	−0.08	−0.6	0.44	0.76	36.6	0.16	−0.16	−2.83	2.5	0.91	36.58	0.13	0.16	-	−0.09	0.27	0.74
	HDL-C	44	4	−0.12	−0.32	0.09	0.27	1272.13	1.59 × 10^−238^	−0.35	−0.98	0.28	0.28	1254.88	1.22 × 10^−235^	<1 × 10^−4^	0.7563	−0.13	0.05	0.02
	LDL-C	44	4	0.44	0.05	0.82	0.03	3784.41	-	1	−0.17	2.18	0.1	3698.32	-	<1 × 10^−4^	<1 × 10^−4^	0.08	0.07	0.25
	TC	45	4	0.33	0.03	0.63	0.03	652	3.30 × 10^−109^	0.76	−0.19	1.7	0.12	638.86	3.95 × 10^−107^	<1 × 10^−4^	<1 × 10^−4^	0.11	0.08	0.16
	TG	44	4	0.19	0.03	0.34	0.02	663.34	4.11 × 10^−112^	0.37	−0.1	0.84	0.13	653.47	1.06 × 10^−110^	<1 × 10^−4^	0.1338	0.15	0.02	0
	T2D	6	5	0.1	0.01	0.71	0.02	80.3	-	0.19	6.56 × 10^−5^	560.9	0.69	79.55	-	2.00 × 10^−4^	<1 × 10^−4^	−1.68	0.63	0.01
	Stroke	0	-	-	-	-	-	-	-	-	-	-	-	-	-	-	-	-	-	-
	* T2D	5	5	0.47	0.3	0.75	0.001	3.51	0.48	0.18	0.04	0.72	0.02	1.42	0.7	0.4343	-	−0.82	0.29	0.004
	CHD	44	4	1.23	0.92	1.64	0.2	95.94	6.50 × 10^−6^	1.12	0.44	2.87	0.76	95.85	4.30 × 10^−6^	<1 × 10^−4^	0.2603	0.17	0.12	0.16
**Proteins**	FG	24	5	−0.12	−0.32	0.09	0.26	156.55	-	0.24	−0.44	0.92	0.49	148.81	-	<1 × 10^−4^	0.8797	−0.1	0.08	0.24
	2h glucose	24	5	0.14	−0.58	0.86	0.7	52.9	3.78 × 10^−4^	−0.98	-3.43	1.47	0.43	51.59	3.56 × 10^−4^	3.00 × 10^−4^	0.0805	−0.07	0.32	0.83
	HDL-C	38	5	−0.18	−0.34	−0.01	0.03	656.82	1.81 × 10^−114^	−0.28	−0.7	0.15	0.2	653.96	1.63 × 10^−114^	<1 × 10^−4^	<1 × 10^−4^	−0.07	0.05	0.14
	LDL-C	38	5	−0.19	−0.36	−0.03	0.02	564.08	1.75 × 10^−95^	−0.47	−0.9	−0.05	0.03	540.64	2.63 × 10^−91^	<1 × 10^−4^	<1 × 10^−4^	0.1	0.06	0.09
	TC	38	5	−0.19	−0.41	0.03	0.09	275.49	8.63 × 10^−38^	−0.53	−1.06	0.01	0.05	262.63	8.45 × 10^−36^	<1 × 10^−4^	0.0876	−0.14	0.08	0.09
	TG	38	5	0.04	−0.26	0.34	0.78	1927.21	-	−0.37	−1.13	0.38	0.33	1993.84	-	<1 × 10^−4^	0.023	−0.07	0.06	0.24
	T2D	4	6	1.78	0.03	105.08	0.78	219.15	-	0.01	4.81 × 10^−27^	1.57 × 10^22^	0.88	215.3	-	<1 × 10^−4^	-	−0.76	1.63	0.64
	Stroke	38	5	0.93	0.72	1.19	0.68	58.95	0.01	0.84	0.43	1.66	0.51	58.79	0.01	0.01	0.13	−0.03	0.12	0.81
	* T2D	4	6	0.61	0.07	5.38	0.66	87.4	-	9.77	1.19 × 10^−14^	7.99 × 10^15^	0.91	86.4	-	<1 × 10^−4^	<1 × 10^−4^	−0.44	1.07	0.68
	CHD	38	5	1.09	0.83	1.43	0.46	66.19	2.20 × 10^−3^	0.95	0.46	1.96	0.82			1.90 × 10^−3^	-	0.07	0.11	0.53

For T2D, Stroke, * T2D and CHD outcomes the effect estimate correspond to Odds ratio (OR); * adjusted for BMI; ** Wald ratio method for single SNP; (-) Not possible to estimate; We considered significant if the directions of the estimates by IVW, weighted median (Appendix A) and MR-Egger were directionally consistent with *p* < 0.05, and no significant evidence of pleiotropy tested by MR-PRESSO (*p* > 0.05). F statistics (median) for the strength of correlation between instrument and exposure. IVW: inverse variance weighted; MR-RAPS: Robust adjusted profile score; MR-PRESSO: Pleiotropy residual sum and outlier.T2D: Type 2 diabetes; CHD: Coronary heart disease; FG: fasting glucose; 2 h glucose: two-hour glucose; HDL-C: high-density lipoprotein; LDL-C: low-density lipoprotein; TG: triglycerides; TC: total cholesterol. F-statistic corresponds to the median.

## Data Availability

The exposures and outcomes GWAS summary statistics are available in: CAD (URL: http://www.cardiogramplusc4d.org/data-downloads/, accessed 1 July 2021) [27]. Stroke (URL: https://megastroke.org/download.html/, accessed 1 July 2021) [28]. T2D (URL: https://www.diagram-consortium.org/downloads.html/, accessed 1 July 2021 [29]). Fasting and 2h glucose (URL: https://www.magicinvestigators.org/downloads/ accessed 1 July 2021 [30,31]). HDL-C, LDL-C, TG and TC (URL: https://gwas.mrcieu.ac.uk/datasets/, accessed 1 July 2021 [39]). The individual level data from VHU are not publicly available due privacy and confidentiality constraints of Swedish regulation, but data are available from the Department of Biobank Research, Umeå University, upon reasonable request.

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
