# Peer review of "Estimating the Direct Effect between Dietary Macronutrients and Cardiometabolic Disease, Accounting for Mediation by Adiposity and Physical Activity"

_nutrients, 2022, doi:10.3390/nu14061218_

Round 1

Reviewer 1 Report

This is a very well designed, performed and written study, with powerfull methodology. The limitations of study are well described. I have no comments.

Author Response

Comment R1.1

This is a very well designed, performed and written study, with powerfull methodology. The limitations of study are well described. I have no comments.

Response R1.1:

We thank the reviewer for her/his kind remarks.

Reviewer 2 Report

Overall

Thank you for this interesting paper. My main comment relates to the use of the term ‘causal’ which presents issues for me, given the type of study you are using (observational) and the quality of the dietary data (self-reported FFQ). While you have explained the mathematical connections between direct dietary variables impacting on disease risks and indirect variables, this is insufficient to get around the practical issue of dietary and lifestyle confounding. You state that macronutrients are rarely eaten alone and, so, it is highly likely that the carbohydrates, sugars and fats are part of complex dietary patterns where it is impossible from a physiological perspective to pick out the isolated metabolic effects of one nutrient. The only way around this problem is to change ‘causal’ to ‘direct association’, ‘direct protective/detrimental signal’ or a similar terminology throughout the paper. ‘Causal’ in my view implies a direct physiological effect on disease risk, such as smoking and lung cancer or folic acid and neural tube defects, which may not be the case in your dataset.

I am concerned that Figure 2 is unreadable even when I blow up the size to 150%. Since you don’t discuss the micronutrients or fatty acids anywhere else in the paper, can these be removed from the figure, giving you more room to increase the size of the other text? I am interested why DHA and EPA seem to have a negative impact on health when I would expect the opposite (Figure 2). You don’t discuss this in the paper so either remove from the Figure or add some thoughts on it.

As the Figures are not easy to read, and the supplementary data require additional downloads, it would be more helpful for readers of your paper to have a simple table showing which nutrients were associated (P<0.05) with disease risks. Can this be added?

As it is unclear to me from reading Figure 2 which nutrients are statistically associated with disease risk, I would like to check that you have ensured that all significant findings are included in your Discussion. There could be a tendency to focus on those with expected directions of association (e.g., plant protein and fibre being positive) and ignore those with unexpected directions of association – for example sucrose seems to be a positive nutrient while vitamin A & D are negative, although I’m not able to tell whether these are statistically significant.

Abbreviations – there are a lot of these throughout and some are not written in full the first time (e.g., GWAS, DAG), which is good practice. I recommend a list of abbreviations at the end of the article that readers can refer to.

Minor comments

Line 65: who performed the ‘extensive health examination’? Medical personal or researchers? Were they trained to do so?

Line 67: FFQ is not an appropriate abbreviation for ‘food frequency intake’ as it generally means ‘food frequency questionnaire’. Please change

Line 73: did ‘exposure data’ include dietary supplements?

Line 310: should ‘likley’ be ‘likely’?
